# Relational Neural Expectation Maximization: Unsupervised Discovery of Objects and their Interactions

**Sjoerd van Steenkiste**
Swiss AI Lab IDSIA, SUPSI, USI
Lugano, Switzerland
sjoerd@idsia.ch

**Michael Chang** *
UC Berkeley
Berkeley, United States
mbchang@berkeley.edu

**Klaus Greff**
Swiss AI Lab IDSIA, SUPSI, USI
Lugano, Switzerland
klaus@idsia.ch

**Jürgen Schmidhuber**
Swiss AI Lab IDSIA, SUPSI, USI
Lugano, Switzerland
juergen@idsia.ch

## Abstract

Common-sense physical reasoning is an essential ingredient for any intelligent agent operating in the real-world. For example, it can be used to simulate the environment, or to infer the state of parts of the world that are currently unobserved. In order to match real-world conditions this causal knowledge must be learned without access to supervised data. To address this problem we present a novel method that learns to discover objects and model their physical interactions from raw visual images in a purely *unsupervised* fashion. It incorporates prior knowledge about the compositional nature of human perception to factor interactions between object-pairs and learn efficiently. On videos of bouncing balls we show the superior modelling capabilities of our method compared to other unsupervised neural approaches that do not incorporate such prior knowledge. We demonstrate its ability to handle occlusion and show that it can extrapolate learned knowledge to scenes with different numbers of objects.

## 1 Introduction

Humans rely on common-sense physical reasoning to solve many everyday physics-related tasks (Lake et al., 2016). For example, it enables them to foresee the consequences of their actions (simulation), or to infer the state of parts of the world that are currently unobserved. This *causal* understanding is an essential ingredient for any intelligent agent that is to operate within the world.

Common-sense physical reasoning is facilitated by the discovery and representation of objects (a *core* domain of human cognition (Spelke & Kinzler, 2007)) that serve as primitives of a compositional system. They allow humans to decompose a complex visual scene into distinct parts, describe relations between them and reason about their dynamics as well as the consequences of their interactions (Battaglia et al., 2013; Lake et al., 2016; Ullman et al., 2017).

The most successful machine learning approaches to common-sense physical reasoning incorporate such prior knowledge in their design. They maintain explicit object representations, which allow for general physical dynamics to be learned between object pairs in a compositional manner (Battaglia et al., 2016; Chang et al., 2016; Watters et al., 2017). However, in these approaches learning is *supervised*, as it relies on object-representations from external sources (e.g. a physics simulator) that are typically unavailable in real-world scenarios.

Neural approaches that learn to directly model motion or physical interactions in pixel space offer an alternative solution (Srivastava et al., 2015; Sutskever et al., 2009). However, while unsupervised,

---

*Work performed while at IDSIA.

these methods suffer from a lack compositionality at the representational level of objects. This prevents such end-to-end neural approaches from efficiently learning functions that operate on multiple entities and generalize in a human-like way (c.f. Battaglia et al. (2013); Lake et al. (2016); Santoro et al. (2017), but see Perez et al. (2017)).

In this work we propose *Relational N-EM* (R-NEM), a novel approach to common-sense physical reasoning that learns physical interactions between objects from raw visual images in a purely *unsupervised* fashion. At its core is *Neural Expectation Maximization* (N-EM; Greff et al., 2017), a method that allows for the discovery of compositional object-representations, yet is unable to model interactions between objects. Therefore, we endow N-EM with a relational mechanism inspired by previous work (Battaglia et al., 2016; Chang et al., 2016; Santoro et al., 2017), enabling it to factor interactions between object-pairs, learn efficiently, and generalize to visual scenes with a varying number of objects without re-training.

## 2 METHOD

Our goal is to learn common-sense physical reasoning in a purely unsupervised fashion directly from visual observations. We have argued that in order to solve this problem we need to exploit the compositional structure of a visual scene. Conventional unsupervised representation learning approaches (eg. VAEs Kingma & Welling (2013); GANs Goodfellow et al. (2014)) learn a single distributed representation that *superimposes* information about the input, without imposing any structure regarding objects or other low-level primitives. These monolithic representations can not factorize physical interactions between pairs of objects and therefore lack an essential inductive bias to learn these efficiently. Hence, we require an alternative approach that can discover objects representations as primitives of a visual scene in an unsupervised fashion.

One such approach is *Neural Expectation Maximization* (N-EM; Greff et al. (2017)), which learns a separate distributed representation for each object described in terms of the same features through an iterative process of perceptual grouping and representation learning. The compositional nature of these representations enable us to formulate *Relational N-EM* (R-NEM): a novel *unsupervised* approach to common-sense physical reasoning that combines N-EM (Section 2.1) with an interaction function that models relations between objects efficiently (Section 2.2).

### 2.1 NEURAL EXPECTATION MAXIMIZATION

*Neural Expectation Maximization* (N-EM; Greff et al. (2017)) is a differentiable clustering method that learns a representation of a visual scene composed of primitive object representations. These representations adhere to many useful properties of a symbolic representation of objects, and can therefore be used as primitives of a compositional system (Hummel et al., 2004). They are described in the same format and each contain only information about the object in the visual scene that they correspond to. Together, they form a representation of a visual scene composed of objects that is learned in an unsupervised way, which therefore serves as a starting point for our approach.

The goal of N-EM is to group pixels in the input that belong to the same object (perceptual grouping) and capture this information efficiently in a distributed representation $\boldsymbol{\theta}_k$ for each object. At a high-level, the idea is that if we were to have access to the family of distributions $P(\boldsymbol{x}|\boldsymbol{\theta}_k)$ (a statistical model of images given object representations $\boldsymbol{\theta}_k$) then we can formalize our objective as inference in a mixture of these distributions. By using *Expectation Maximization* (EM; Dempster et al., 1977) to compute a Maximum Likelihood Estimate (MLE) of the parameters of this mixture $(\boldsymbol{\theta}_1, \ldots, \boldsymbol{\theta}_K)$, we obtain a grouping (clustering) of the pixels to each object (component) and their corresponding representation. In reality we do not have access to $P(\boldsymbol{x}|\boldsymbol{\theta}_k)$, which N-EM learns instead by parameterizing the mixture with a neural network and back-propagating through the iterations of the unrolled generalized EM procedure.

Following Greff et al. (2017), we model each image $\boldsymbol{x} \in \mathbb{R}^D$ as a spatial mixture of $K$ components parameterized by vectors $\boldsymbol{\theta}_1, \ldots, \boldsymbol{\theta}_K \in \mathbb{R}^M$. A neural network $f_\phi$ is used to transform these representations $\boldsymbol{\theta}_k$ into parameters $\psi_{i,k} = f_\phi(\boldsymbol{\theta}_k)_i$ for separate pixel-wise distributions. A set of binary latent variables $\boldsymbol{\mathcal{Z}} \in [0,1]^{D \times K}$ encodes the unknown true pixel assignments, such that $z_{i,k} = 1$ iff pixel $i$ was generated by component $k$. The full likelihood for $\boldsymbol{x}$ given $\boldsymbol{\theta} = (\boldsymbol{\theta}_1, \ldots, \boldsymbol{\theta}_K)$

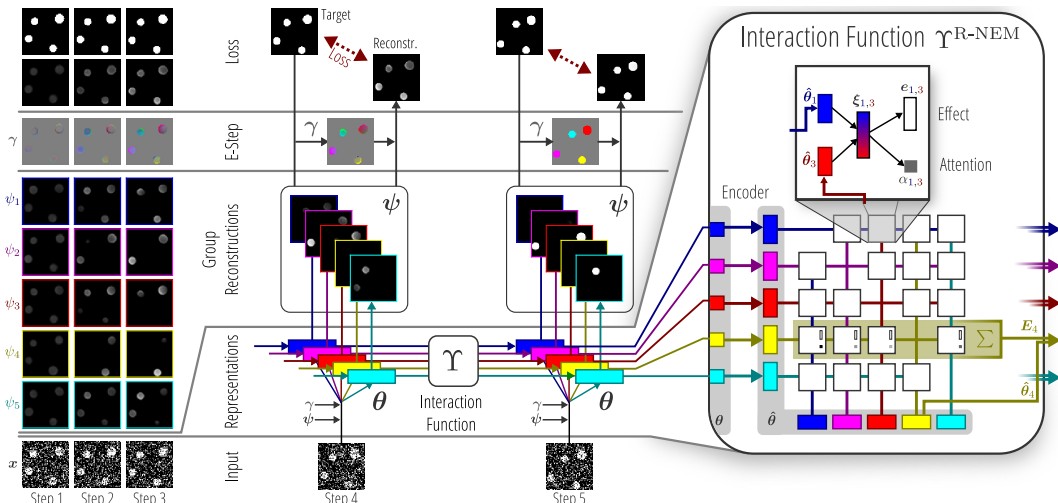

Figure 1: Illustration of the different computational aspects of R-NEM when applied to a sequence of images of bouncing balls. Note that $\gamma, \psi$ at the *Representations* level correspond to the $\gamma$ (*E-step*), $\psi$ (*Group Reconstructions*) from the previous time-step. Different colors correspond to different cluster components (object representations).The right side shows a computational overview of $\Upsilon^{\text{R-NEM}}$, a function that computes the pair-wise interactions between the object representations.

is given by:

$$P(\boldsymbol{x}|\boldsymbol{\theta}) = \prod_{i=1}^{D} \sum_{\boldsymbol{z}_i} P(x_i, \boldsymbol{z}_i|\boldsymbol{\psi}_i) = \prod_{i=1}^{D} \sum_{k=1}^{K} P(z_{i,k} = 1)P(x_i|\psi_{i,k}, z_{i,k} = 1). \tag{1}$$

If $f_\phi$ has learned a statistical model of images given object representations $\boldsymbol{\theta}_k$, then we can compute the object representations for a given image $\boldsymbol{x}$ by maximizing $P(\boldsymbol{x}|\boldsymbol{\theta})$. Marginalization over $\boldsymbol{z}$ complicates this process, thus we use generalized EM to maximize the following lowerbound instead:

$$\mathcal{Q}(\boldsymbol{\theta}, \boldsymbol{\theta}^{\text{old}}) = \sum_{\mathbf{z}} P(\mathbf{z}|\boldsymbol{x}, \boldsymbol{\psi}^{\text{old}}) \log P(\boldsymbol{x}, \mathbf{z}|\boldsymbol{\psi}). \tag{2}$$

Each iteration of generalized EM consists of two steps: the *E-step* computes a new estimate of the posterior probability distribution over the latent variables $\gamma_{i,k} := P(z_{i,k} = 1|x_i, \psi_i^{\text{old}})$ given $\boldsymbol{\theta}^{\text{old}}$ from the previous iteration. It yields a new soft-assignment of the pixels to the components (clusters), based on how accurately they model $\boldsymbol{x}$. The generalized *M-step* updates $\boldsymbol{\theta}^{\text{old}}$ by taking a gradient ascent step on (2), using the previously computed soft-assignments: $\boldsymbol{\theta}_k^{\text{new}} = \boldsymbol{\theta}_k^{\text{old}} + \eta \cdot \partial\mathcal{Q}/\partial\boldsymbol{\theta}_k$.[1]

The unrolled computational graph of the generalized EM steps is differentiable, which provides a means to train $f_\phi$ to implement a statistical model of images given object representations. Using back-propagation through time (eg. Werbos (1988); Williams (1989)) we train $f_\phi$ to minimize the following loss:

$$L(\boldsymbol{x}) = -\sum_{i=1}^{D} \sum_{k=1}^{K} \underbrace{\gamma_{i,k} \log P(x_i, z_{i,k}|\psi_{i,k})}_{\text{intra-cluster loss}} - \underbrace{(1 - \gamma_{i,k})D_{KL}[P(x_i)||P(x_i|\psi_{i,k}, z_{i,k})]}_{\text{inter-cluster loss}}. \tag{3}$$

The intra-cluster term is identical to (2), which credits each component for accurately representing pixels that have been assigned to it. The inter-cluster term ensures that each representation only captures the information about the pixels that have been assigned to it.

---

[1] We can not compute $\text{argmax}_\theta \mathcal{Q}(\boldsymbol{\theta}, \boldsymbol{\theta}^{\text{old}})$ analytically, due to non-linearity of $f_\phi$.

A more powerful variant of N-EM can be obtained (RNN-EM) by substituting the generalized M-step with a recurrent neural network having hidden state $\boldsymbol{\theta}_k$. In this case, the entirety of $f_\phi$ consists of a recurrent encoder-decoder architecture that receives $\boldsymbol{\gamma}_k(\boldsymbol{x} - \boldsymbol{\psi}_k)$ as input at each step.

The learning objective in (3) is prone to trivial solutions in case of overcapacity, which could prevent the network from modelling the statistical regularities in the data that correspond to objects. By adding noise to the input image or reducing $\boldsymbol{\theta}$ in dimensionality we can guide learning to avert this. Moreover, in the case of RNN-EM one can evaluate (3) at the following time-step (*predictive coding*) to encourage learning of object representations and their corresponding dynamics. One intuitive interpretation of using denoising or next-step prediction as part of the training objective is to guide the network to learn about essential properties of objects, in this case those that correspond to the Gestalt Principles of *prägnanz* and *common fate* (Hatfield & Epstein, 1985).

## 2.2 Relational Neural Expectation Maximization

RNN-EM (unlike N-EM) is able to capture the dynamics of individual objects through a parametrized recurrent connection that operates on the object representation $\boldsymbol{\theta_k}$ across consecutive time-steps. However, the relations and interactions that take place *between* objects can not be captured in this way. In order to overcome this shortcoming we propose *Relational N-EM* (R-NEM), which adds relational structure to the recurrence to model interactions between objects without violating key properties of the learned object representations.

Consider a generalized form of the standard RNN-EM dynamics equation, which computes the object representation $\boldsymbol{\theta}_k$ at time $t$ as a function of all object representations $\boldsymbol{\theta} := [\boldsymbol{\theta}_1, \ldots, \boldsymbol{\theta}_K]$ at the previous time-step through an *interaction function* $\Upsilon$:

$$\boldsymbol{\theta}_k^{(t)} = \text{RNN}(\tilde{\boldsymbol{x}}^{(t)}, \Upsilon_k(\boldsymbol{\theta}^{(t-1)})) := \sigma(\boldsymbol{W} \cdot \tilde{\boldsymbol{x}}^{(t)} + \boldsymbol{R} \cdot \Upsilon_k(\boldsymbol{\theta}^{(t-1)})). \tag{4}$$

Here $\boldsymbol{W}, \boldsymbol{R}$ are weight matrices, $\sigma$ is the sigmoid activation function, and $\tilde{\boldsymbol{x}}^{(t)}$ is the input to the recurrent model at time $t$ (possibly transformed by an encoder). When $\Upsilon_k^{\text{RNN-EM}}(\boldsymbol{\theta}) := \boldsymbol{\theta}_k$, this dynamics model coincides with a standard RNN update rule, thereby recovering the original RNN-EM formulation.

The inductive bias incorporated in $\Upsilon$ reflects the modeling assumptions about the interactions between objects in the environment, and therefore the nature of $\boldsymbol{\theta}_k$'s interdependence. If $\Upsilon$ incorporates the assumption that no interaction takes place between objects, then the $\boldsymbol{\theta}_k$'s are fully independent and we recover $\Upsilon^{\text{RNN-EM}}$. On the other hand, if we do assume that interactions among objects take place, but assume very little about the structure of the interdependence between the $\boldsymbol{\theta}_k$'s, then we forfeit useful properties of $\boldsymbol{\theta}_k$ such as compositionality. For example, if $\Upsilon := \text{MLP}(\boldsymbol{\theta})$ we can no longer extrapolate learned knowledge to environments with more or fewer than $K$ objects and lose overall data efficiency (Santoro et al., 2017). Instead, we can make efficient use of compositionality among the learned object representations $\boldsymbol{\theta}_k$ to incorporate general but guiding constraints on how these may influence one another (Battaglia et al., 2016; Chang et al., 2016). In doing so we constrain $\Upsilon$ to capture interdependence between $\boldsymbol{\theta}_k$'s in a compositional manner that enables physical dynamics to be learned efficiently, and allow for learned dynamics to be extrapolated to a variable number of objects.

We propose a parametrized interaction function $\Upsilon^{\text{R-NEM}}$ that incorporates these modeling assumptions and updates $\boldsymbol{\theta}_k$ based on the pairwise effects of the objects $i \neq k$ on $k$:

$$\Upsilon_k^{\text{R-NEM}}(\boldsymbol{\theta}) = [\hat{\boldsymbol{\theta}}_k; \boldsymbol{E}_k] \quad \text{with} \quad \hat{\boldsymbol{\theta}}_k = \text{MLP}^{enc}(\boldsymbol{\theta}_k) \ , \quad \boldsymbol{E}_k = \sum_{i \neq k} \alpha_{k,i} \cdot \boldsymbol{e}_{k,i}$$

$$\alpha_{k,i} = \text{MLP}^{att}(\boldsymbol{\xi}_{k,i}) \ , \quad \boldsymbol{e}_{k,i} = \text{MLP}^{eff}(\boldsymbol{\xi}_{k,i}) \ , \quad \boldsymbol{\xi}_{k,i} = \text{MLP}^{emb}([\hat{\boldsymbol{\theta}}_k; \hat{\boldsymbol{\theta}}_i]) \tag{5}$$

where $[\cdot; \cdot]$ is the concatenation operator and $\text{MLP}^{(\cdot)}$ corresponds to a multi-layer perceptron. First, each $\boldsymbol{\theta}_i$ is transformed using $\text{MLP}^{enc}$ to obtain $\hat{\boldsymbol{\theta}}_i$, which enables information that is relevant for the object dynamics to be made more explicit in the representation. Next, each pair $(\hat{\boldsymbol{\theta}}_k, \hat{\boldsymbol{\theta}}_i)$ is concatenated and processed by $\text{MLP}^{emb}$, which computes a shared embedding $\boldsymbol{\xi}_{k,i}$ that encodes the interaction between object $k$ and object $i$. Notice that we opt for a clear separation between the *focus* object $k$ and the *context* object $i$ as in previous work (Chang et al., 2016). From $\boldsymbol{\xi}_{k,i}$ we compute $\boldsymbol{e}_{k,i}$: the effect of object $i$ on object $k$; and an attention coefficient $\alpha_{k,i}$ that encodes whether interaction

between object $i$ and object $k$ takes place. These attention coefficients (Bahdanau et al., 2014; Xu et al., 2015) help to select relevant context objects, and can be seen as a more flexible unsupervised replacement of the distance based heuristic that was used in previous work (Chang et al., 2016). Finally, we compute the total effect of $\boldsymbol{\theta}_{i \neq k}$ on $\boldsymbol{\theta}_k$ as a weighted sum of the effects multiplied by their attention coefficient. A visual overview of $\Upsilon^{\text{R-NEM}}$ can be seen on the right side of Figure 1.

## 3   RELATED WORK

Machine learning approaches to common-sense physical reasoning can roughly be divided in two groups: symbolic approaches and approaches that perform state-to-state prediction. The former group performs inference over the parameters of a symbolic physics engine (Battaglia et al., 2013; Ullman et al., 2017; Wu et al., 2015), which restricts them to synthetic environments. The latter group employs machine learning methods to make state-to-state predictions, often describing the state of a system as a set of compact object-descriptions that are either used as an input to the system (Battaglia et al., 2016; Chang et al., 2016; Fragkiadaki et al., 2015; Grzeszczuk et al., 1998) or for training purposes (Watters et al., 2017). By incorporating information (eg. position, velocity) about objects these methods have achieved excellent generalization and simulation capabilities. Purely unsupervised approaches for state-to-state prediction (Agrawal et al., 2016; Lerer et al., 2016; Michalski et al., 2014; Sutskever et al., 2009) that use raw visual inputs as state-descriptions have yet to rival these capabilities. Our method is a purely unsupervised state-to-state prediction method that operates in pixel space, taking a first step towards unsupervised learning of common-sense reasoning in real-world environments.

The proposed interaction function $\Upsilon^{\text{R-NEM}}$ can be seen as a type of Message Passing Neural Network (MPNN; Gilmer et al. (2017)) that incorporates a variant of neighborhood attention (Duan et al., 2017). In light of other recent work (Zaheer et al., 2017) it can be seen as a permutation equivariant set function.

R-NEM relies on N-EM (Greff et al., 2017) to discover a compositional object representation from raw visual inputs. A closely related approach to N-EM is the TAG framework (Greff et al., 2016), which utilizes a similar mechanism to perform inference over group representations, but in addition performs inference over the group assignments. In recent work TAG was combined with a recurrent ladder network (Ilin et al., 2017) to obtain a powerful model (RTagger) that can be applied to sequential data. However, the lack of a single compact representation that captures all information about a group (object) makes a compositional treatment of physical interactions more difficult. Other unsupervised approaches rely on attention to group together parts of the visual scene corresponding to objects (Eslami et al., 2016; Gregor et al., 2015). These approaches suffer from a similar problem in that their sequential nature prevents a coherent object representation to take shape.

Other related work have also taken steps towards combining the learnability of neural networks with the compositionality of symbolic programs in modeling physics (Battaglia et al., 2016; Chang et al., 2016), playing games (Denil et al., 2017; Kansky et al., 2017), learning algorithms (Bošnjak et al., 2017; Cai et al., 2017; Li et al., 2016; Reed & De Freitas, 2015), visual understanding (Ellis et al., 2017; Johnson et al., 2017), and natural language processing (Andreas et al., 2016; Hu et al., 2017).

## 4   EXPERIMENTS

In this section we evaluate R-NEM on three different physical reasoning tasks that each vary in their dynamical and visual complexity: bouncing balls with variable mass, bouncing balls with an invisible curtain and the Arcade Learning Environment (Bellemare et al., 2013). We compare R-NEM to other *unsupervised* neural methods that do not incorporate any inductive biases reflecting real-world dynamics and show that these are indeed beneficial.[2]

All experiments use ADAM (Kingma & Ba, 2014) with default parameters, on 50K train + 10K validation + 10K test sequences and early stopping with a patience of 10 epochs. For each of $\text{MLP}^{enc,emb,eff}$ we used a unique single layer neural network with 250 *rectified linear* units. For $\text{MLP}^{att}$ we used a two-layer neural network: 100 *tanh* units followed by a single *sigmoid* unit. A detailed overview of the experimental setup can be found in Appendix A.

---

[2]Code is available at `https://github.com/sjoerdvansteenkiste/Relational-NEM`.

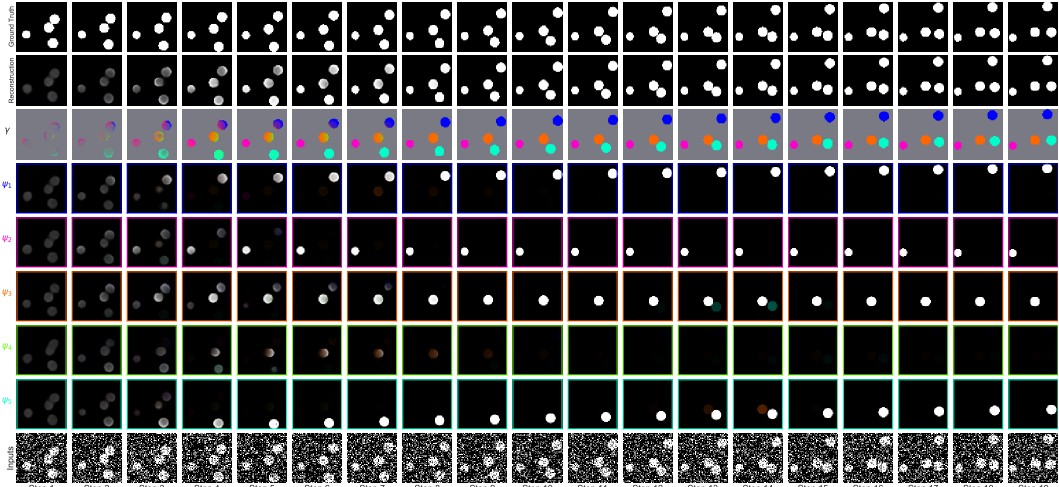

Figure 2: R-NEM applied to a sequence of 4 bouncing balls. Each column corresponds to a time-step, which coincides with an EM step. At each time-step, R-NEM computes $K = 5$ new representations $\boldsymbol{\theta}_k$ according to (4) (see also *Representations* in Figure 1) from the input $\boldsymbol{x}$ with added noise (bottom row). From each new $\boldsymbol{\theta}_k$ a group reconstruction $\boldsymbol{\psi}_k$ is produced (rows 2-6 from bottom) that predicts the state of the environment at the *next* time-step. Attention coefficients are visualized by overlaying a colored reconstruction of a context object on the white reconstruction of the focus object (see *Attention* in Section 4). Based on the prediction accuracy of $\boldsymbol{\psi}$, the *E-step* (see Figure 1) computes new soft-assignments $\boldsymbol{\gamma}$ (row 7 from bottom), visualized by coloring each pixel $i$ according to their distribution over components $\boldsymbol{\gamma}_i$. Row 8 visualizes the total prediction by the network ($\sum_k \boldsymbol{\psi}_k \cdot \boldsymbol{\gamma}_k$) and row 9 the ground-truth sequence at the next time-step.

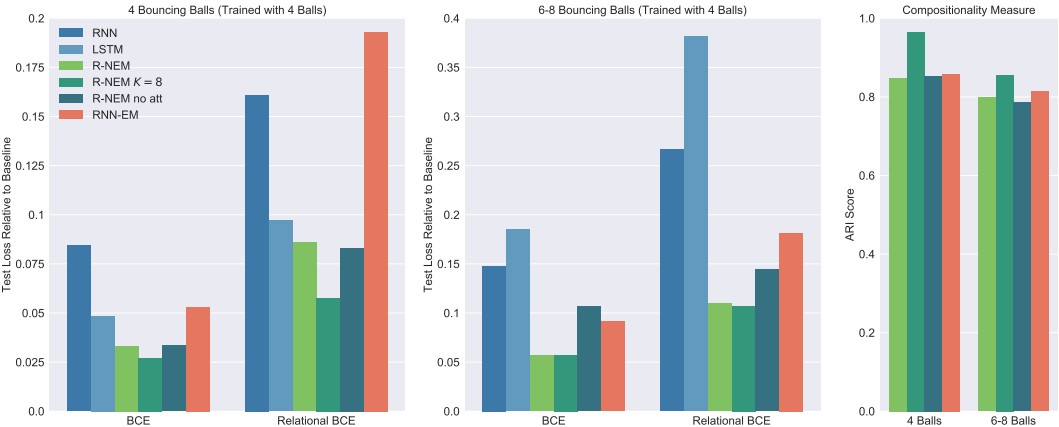

Figure 3: Performance of each method on the *bouncing balls* task. Each method was trained on a dataset with 4 balls, evaluated on a test set with 4 balls (left), and on a test-set with 6-8 balls (middle). The losses are reported relative to the loss of a baseline for each dataset that always predicts the current frame. The ARI score (right) is used to evaluate the degree of compositionality that is achieved.

Figure 4: **Left**: Three sequences of 15 time-steps ground-truth (top), R-NEM (middle), RNN (bottom). The last ten time-steps of the sequences produced by R-NEM and RNN are simulated. **Right**: The BCE loss on the entire test-set for these same time-steps.

**Bouncing Balls**   We study the physical reasoning capabilities of R-NEM on the *bouncing balls* task, a standard environment to evaluate physical reasoning capabilities that exhibits low visual complexity and complex non-linear physical dynamics.[3] We train R-NEM on sequences of $64 \times 64$ binary images over 30 time-steps that contain four bouncing balls with different masses corresponding to their radii. The balls are initialized with random initial positions, masses and velocities. Balls bounce elastically against each other and the image window.

**Qualitative Evaluation**   Figure 1 presents a qualitative evaluation of R-NEM on the bouncing balls task. After 10 time-steps it can be observed that the pixels that belong to each of the balls are grouped together and assigned to a unique component (with a saturated color); and that the background (colored grey) has been divided among all components (resulting in a grey coloring). This indicates that the representation $\boldsymbol{\theta}_k$ from which each component produces the group reconstruction $\boldsymbol{\psi}_k$ does indeed only contain information about a unique object, such that together the $\boldsymbol{\theta}_k$'s yield a compositional object representation of the scene. The total reconstruction (that combines the group reconstructions and the soft-assignments) displays an accurate reconstruction of the input sequence at the next time-step, indicating that R-NEM has learned to model the dynamics of bouncing balls.

**Comparison**   We compare the modelling capabilities of R-NEM to an RNN, LSTM (Gers et al., 1999; Hochreiter & Schmidhuber, 1997) and RNN-EM in terms of the Binomial Cross-Entropy (BCE) loss between the predicted image and the ground-truth image of the last frame,[4] as well as the *relational* BCE that only takes into account objects that currently take part in collision. Unless specified we use $K = 5$.

On a test-set with sequences containing four balls we observe that R-NEM produces markedly lower losses when compared to all other methods (left plot in Figure 3). Moreover, in order to validate that each component captures only a single ball (and thus compositionality is achieved), we report the Adjusted Rand Index (ARI; Hubert & Arabie (1985)) score between the soft-assignments $\gamma$ and the ground-truth assignment of pixels to objects. In the left column of the ARI plot (right side in Figure 3) we find that R-NEM achieves an ARI score of 0.8, meaning that in roughly $80\%$ of the cases each ball is modeled by a single component. This suggests that a compositional object representation is achieved for most of the sequences. Together these observations are in line with our qualitative evaluation and validate that incorporating real world priors is greatly beneficial (comparing to RNN, LSTM) and that $\Upsilon^{\text{R-NEM}}$ enables interactions to be modelled more accurately compared to RNN-EM in terms of the relational BCE.

Similar to Greff et al. (2017) we find that further increasing the number of components during training (leaving additional groups empty) increases the quality of the grouping, see R-NEM $K = 8$ in Figure 3. In addition we observe that the loss (in particular the relational BCE) is reduced further, which matches our hypothesis that compositional object representations are greatly beneficial for modelling physical interactions.

**Extrapolating learned knowledge**   We use a test-set with sequences containing 6-8 balls to evaluate the ability of each method to *extrapolate* their learned knowledge about physical interactions between four balls to environments with more balls. We use $K = 8$ when evaluating R-NEM and

---

[3]Videos are available at `https://sites.google.com/view/r-nem-gifs/`.

[4]Since the E-step in R-NEM and RNN-EM utilizes the ground-truth for reconstruction, we substitute it with a simple `max` operator. The resulting loss serves as an *upperbound* to the true BCE loss.

RNN-EM on this test-set in order to accommodate the increased number of objects. As can be seen from the middle plot in Figure 3, R-NEM again greatly outperforms all other methods. Notice that, since we report the loss relative to a baseline, we roughly factor out the increased complexity of the task. Perfect extrapolation of the learned knowledge would therefore amount to *no change* in relative performance. In contrast, we observe far worse performance for the LSTM (relative to the baseline) when evaluated on this dataset with extra balls. It suggests that the gating mechanism of the LSTM has allowed it to learn a sophisticated and overly specialized solution for sequences with four balls that does not generalize to a dataset with 6-8 balls.

R-NEM and RNN-EM scale markedly better to this dataset than LSTM. Although the RNN similarly suffers to a lesser extend from this type of "overfitting", this is most likely due its inability to learn a reasonable solution on sequences of four balls to begin with. Hence, we conclude that the superior extrapolation capabilities of RNN-EM and R-NEM are inherent to their ability to factor a scene in terms of permutation invariant object representations (see right side of the right plot in Figure 3).

**Attention**    Further insight in the role of the attention mechanism can be gained by visualizing the attention coefficients, as is done in Figure 2. For each component $k$ we draw $\alpha_{k,i} * \psi_i$ on top of the reconstruction $\psi_k$, colored according to the color of component $i$. These correspond to the colored balls (that are for example seen in time-steps 13, 14), which indicate whether component $k$ took information about component $i$ into account when computing the new state (recall (5)). It can be observed that the attention coefficient $\alpha_{k,i}$ becomes non-zero whenever collision takes place, such that a colored ball lights up in the *following* time-steps. The attention mechanism learned by R-NEM thus assumes the role of the distance-based heuristic in previous work (Chang et al., 2016), matching our own intuitions of how this mechanism would best be utilized.

A quantitative evaluation of the attention mechanism is obtained by comparing R-NEM to a variant of itself that does not incorporate attention (*R-NEM no att*). Figure 3 shows that both methods perform equally well on the regular test set (4 balls), but that *R-NEM no att* performs worse at extrapolating from its learned knowledge (6-8 balls). A likely reason for this behavior is that the range of the sum in (5) changes with $K$. Thus, when extrapolating to an environment with more balls the total sum may exceed previous boundaries and impede learned dynamics.

**Simulation**    Once a scene has been accurately modelled, R-NEM can approximately simulate its dynamics through recursive application of (4) for each $\boldsymbol{\theta}_k$.[5] In Figure 4 we compare the simulation capabilities of R-NEM to RNN-EM and an RNN on the bouncing balls environment.[3] On the left it shows for R-NEM and an RNN a sequence with five normal steps followed by 10 simulation steps, as well as the ground-truth sequence. From the last frame in the sequence it can clearly be observed that R-NEM has managed to accurately simulate the environment. Each ball is approximately in the correct place, and the shape of each ball is preserved. The balls simulated by the RNN, on the other hand, deviate substantially from their ground-truth position and their size has increased. In general we find that R-NEM produces mostly very accurate simulations, whereas the RNN consistently fails. Interestingly we found that the cases in which R-NEM frequently fails are those for which a single component models more than one ball. The right side of Figure 4 summarizes the BCE loss for these same time-steps across the *entire* test-set. Although this is a crude measure of simulation performance (since it does not take into account the identity of the balls), we still observe that R-NEM consistently outperforms RNN-EM and an RNN.

**Hidden Factors**    Occlusion is abundant in the real world, and the ability to handle hidden factors is crucial for any physical reasoning system. We therefore evaluate the capability of R-NEM to handle occlusion using a variant of bouncing balls that contain an invisible "curtain." Figure 5 shows that R-NEM accurately models the sequence and can maintain object states, even when confronted with occlusion.[3] For example, note that in step 36 the "blue" ball, is completely occluded and is about to collide with the "orange" ball. In step 38 the ball is accurately predicted to re-appear at the bottom of the curtain (since collision took place) as opposed to the left side of the curtain. This demonstrates that R-NEM has a notion of *object permanence* and implies that it understands a scene on a level beyond pixels: it assigns persistence and identity to the objects.

---

[5]Note that in this case the input to the neural network encoder in component $k$ corresponds to $\boldsymbol{\gamma}_k(\boldsymbol{x}^{(t)} - \boldsymbol{\psi}^{(t-1)})$, such that the output of the encoder $\tilde{\boldsymbol{x}}^{(t)} \approx \boldsymbol{0}$ when $\boldsymbol{\psi}_k^{(t-1)} = \boldsymbol{x}^{(t)}$.

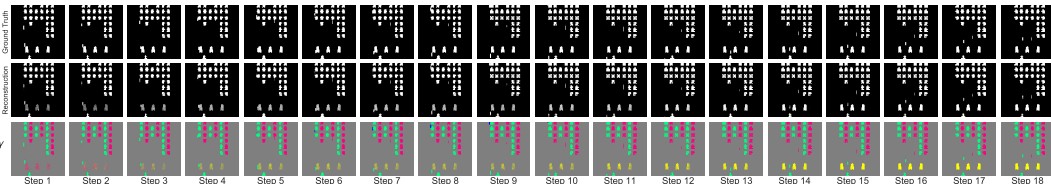

Figure 5: R-NEM applied to a sequence of bouncing balls with an invisible curtain. The ground truth sequence is displayed in the top row, followed by the prediction of R-NEM (middle) and the soft-assignments of pixels to components (bottom). R-NEM models objects, as well as its interactions, even when the object is completely occluded (step 36). Only a subset of the steps is shown.

Figure 6: R-NEM accurately models a sequence of frames obtained by an agent playing Space Invaders. A group no longer corresponds to an object, but instead assumes the role of high-level entities that engage in similar movement patterns.

In terms of test-set performance we find that R-NEM (*BCE:* 46.22, *relational BCE:* 2.33) outperforms an RNN (*BCE:* 94.64, *relational BCE:* 4.14) and an LSTM (*BCE:* 59.32, *relational BCE:* 2.72).

**Space Invaders**   To test the performance of R-NEM in a visually more challenging environment, we train it on sequences of $84 \times 84$ binarized images over 25 time-steps of game-play on Space Invaders from the Arcade Learning Environment (Bellemare et al., 2013).[6] We use $K = 4$ and also feed the action of the agent to the interaction function. Figure 6 confirms that R-NEM is able to accurately model the environment, even though the visual complexity has increased. Notice that these visual scenes comprise a large numbers of (small) primitive objects that behave similarly. Since we trained R-NEM with four components it is unable to group pixels according to individual objects and is forced to consider a different grouping. We find that R-NEM assigns different groups to every other column of aliens together with the spaceship, and to the three large "shields." These groupings seem to be based on movement, which to some degree coincides with their semantic roles of the environment. In other examples (not shown) we also found that R-NEM frequently assigns different groups to every other column of the aliens, and to the three large "shields." Individual bullets and the space ship are less frequently grouped separately, which may have to do with the action-noise of the environment (that controls the movement of the space-ship) and the small size of the bullets at the current resolution that makes them less predictable.

## 5   DISCUSSION AND CONCLUSION

We have argued that the ability to discover and describe a scene in terms of objects provides an essential ingredient for common-sense physical reasoning. This is supported by converging evidence from cognitive science and developmental psychology that intuitive physics and reasoning capabilities are built upon the ability to perceive objects and their interactions (Spelke, 1988; Ullman et al., 2017). The fact that young infants already exhibit this ability, may even suggest an innate bias towards compositionality (Lake et al., 2016; Munakata et al., 1997; Spelke & Kinzler, 2007). Inspired by these observations we have proposed R-NEM, a method that incorporates inductive biases about the existence of objects and interactions, implemented by its clustering objective and interaction function respectively. The specific nature of the objects, and their dynamics and interactions can then be learned efficiently purely from visual observations.

---

[6]Binarization ensures that the color group of the entities on the screen does not give away their grouping.

In our experiments we find that R-NEM indeed captures the (physical) dynamics of various environments more accurately than other methods, and that it exhibits improved generalization to environments with different numbers of objects. It can be used as an approximate simulator of the environment, and to predict movement and collisions of objects, even when they are completely occluded. This demonstrates a notion of object permanence and aligns with evidence that young infants seem to infer that occluded objects move in connected paths and continue to maintain object-specific properties (Spelke, 1990). Moreover, young infants also appear to expect that objects only interact when they come into contact (Spelke, 1990), which is analogous to the behaviour of R-NEM to only attend to other objects when a collision is imminent. In summary, we believe that our method presents an important step towards learning a more human-like model of the world in a completely unsupervised fashion.

Current limitations of our approach revolve around grouping and prediction. What aspects of a scene humans group together typically varies as a function of the task in mind. One may perceive a stack of chairs as a whole if the goal is to move them to another room, or as individual chairs if the goal is to count the number of chairs in the stack. In order to facilitate this *dynamic* grouping one would need to incorporate top-down feedback from an agent into the grouping procedure to deviate from the built-in inductive biases. Another limitation of our approach is the need to incentivize R-NEM to produce useful groupings by injecting noise, or reducing capacity. The former may prevent very small regularities in the input from being detected. Finally the interaction in the E-step among the groups makes it difficult to increase the number of components above ten without causing harmful training instabilities. Due to the multitude of interactions and objectives in R-NEM (and RNN-EM) we find that they are sometimes challenging to train.

In terms of prediction we have implicitly assumed that objects in the environment behave according to rules that can be inferred. This poses a challenge when objects deform in a manner that is difficult to predict (as is the case for objects in Space Invaders due to downsampling). However in practice we find that (once pixels have been grouped together) the masking of the input helps each component in quickly adapting its representation to any unforeseen behaviour across consecutive time steps. Perhaps a more severe limitation of R-NEM (and of RNN-EM in general) is that the second loss term of the outer training objective hinders in modelling more complex varying backgrounds, as the background group would have to predict the "pixel prior" for every other group.

We argue that the ability to engage in common-sense physical reasoning benefits any intelligent agent that needs to operate in a physical environment, which provides exciting future research opportunities. In future work we intend to investigate how top-down feedback from an agent could be incorporated in R-NEM to facilitate dynamic groupings, but also how the compositional representations produced by R-NEM can benefit a reinforcement learner, for example to learn a modular policy that easily generalizes to novel combinations of known objects. Other interactions between a controller C and a model of the world M (implemented by R-NEM) as posed in Schmidhuber (2015) constitute further research directions.

## ACKNOWLEDGEMENTS

The authors wish to thank Tom Griffiths and the anonymous reviewers for helpful comments and constructive feedback. This research was supported by the Swiss National Science Foundation grant 200021_165675/1, the EU project "INPUT" (H2020-ICT-2015 grant no. 687795), and the Zeno Karl Schindler Foundation Summerschool Grant. Chang would like to thank Christiane Born, Sarah Craver, Cinzia Daldini, and the MIT MISTI Program for supporting his stay in Switzerland. We are grateful to NVIDIA Corporation for donating us a DGX-1 as part of the Pioneers of AI Research award, and to IBM for donating a "Minsky" machine.

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

## A    EXPERIMENT DETAILS

In all experiments we train the networks using ADAM (Kingma & Ba, 2014) with default parameters, a batch size of 64 and $50\,000$ train $+ 10\,000$ validation $+ 10\,000$ test inputs. The quality of the learned groupings is evaluated by computing the Adjusted Rand Index (ARI; Hubert & Arabie (1985)) with respect to the ground truth, while ignoring the background and overlap regions (as is consistent with earlier work (Greff et al., 2017)). We use early stopping when the validation loss has not improved for 10 epochs.

### A.1    BOUNCING BALLS

The bouncing balls data is similar to previous work (Sutskever et al., 2009) with a few modifications. The data consists of sequences of $64 \times 64$ binary images over 30 time-steps and balls are randomly sampled from two types: one ball is six times heavier and 1.25 times larger in radius than the other. The balls are initialized with random initial positions and velocities. Balls bounce elastically against each other and the image window.

As in previous work (Greff et al., 2017) we use a convolutional encoder-decoder architecture with a recurrent neural network as bottleneck, that is updated according to (4):

1. $4 \times 4$ conv. 16 ELU. stride 2. layer norm
2. $4 \times 4$ conv. 32 ELU. stride 2. layer norm
3. $4 \times 4$ conv. 64 ELU. stride 2. layer norm
4. fully connected. 512 ELU. layer norm
5. recurrent. 250 Sigmoid. layer norm on the output
6. fully connected. 512 RELU. layer norm
7. fully connected. $8 \times 8 \times 64$ RELU. layer norm
8. $4 \times 4$ reshape 2 nearest-neighbour, conv. 32 RELU. layer norm
9. $4 \times 4$ reshape 2 nearest-neighbour, conv. 16 RELU. layer norm
10. $4 \times 4$ reshape 2 nearest-neighbour, conv. 1 Sigmoid

Instead of using transposed convolutions (to implement the "de-convolution") we first reshape the image using the default nearest-neighbour interpolation followed by a normal convolution in order to avoid frequency artifacts (Odena et al., 2016). Note that we do not add layer norm on the recurrent connection.

At each timestep $t$ we feed $\gamma_{:,k}(\boldsymbol{\psi}_{:,k}^{(t-1)} - \hat{\boldsymbol{x}}^{(t)})$ as input to the network, where $\tilde{\boldsymbol{x}}$ is the input with added bitflip noise ($p = 0.2$). Consistent with earlier work (Greff et al., 2017) R-NEM is trained with a next-step prediction objective, the prior for each pixel in the data is set to a Bernoulli distribution with $p = 0$, and we prevent conflicting gradient updates by not back-propagating any gradients through $\gamma$.

The Interaction Function $\boldsymbol{\Upsilon}^{\text{R-NEM}}$network is structured as follows:

- $\text{MLP}^{enc}$: fully connected. 250 RELU. layer norm
- $\text{MLP}^{emb}$: fully connected. 250 RELU. layer norm
- $\text{MLP}^{eff}$: fully connected. 250 RELU. layer norm
- $\text{MLP}^{att}$: fully connected. 100 Tanh. layer norm - fully connected. 1 Sigmoid.

We experimented with deeper architectures, but were unable to observe significant improvement.

**Comparison and Extrapolation**    In the comparison experiment both R-NEM and RNN-EM are trained with $K = 5$ (unless otherwise mentioned), following insights from Greff et al. (2017). On the extrapolation task we adjusted the number of components at test time to $K = 8$.

When comparing to RNN-EM we used $\boldsymbol{\Upsilon} = \boldsymbol{\Upsilon}^{\text{RNN-EM}}$. For comparing to RNN we set $K = 1$, and used $\boldsymbol{\Upsilon} = \boldsymbol{\Upsilon}^{\text{RNN-EM}}$, yielding a standard recurrent autoencoder that receives at each time-step

the difference between the prediction and the noisy ground-truth as input. In case of LSTM, we additionally replace the recurrent layer with an LSTM update. The R-NEM *no att* model is the same as R-NEM, without MLP$^{att}$, such that $\alpha_{:,:} = 1$

**Simulation**   Since the E-step relies on the ground-truth, which was not available for simulation, we used a thresholded version of $\max_k \psi$ at 0.1 (such that everything below becomes 0 and everything above becomes 1) as a replacement in stead.

**Occlusion**   On the occlusion dataset we used three balls with equal mass. The curtain was spawned at a random location for each sequence. We trained R-NEM with $K = 5$.

## A.2   SPACE INVADERS

We used a pre-trained DQN to produce a dataset with sequences of 25 time-steps. The DQN receives a stack of four frames as input and we recorded every first frame of this stack. These frames were first pre-processed as in Mnih et al. (2013) and then thresholded at 0.0001 to obtain binary images.

Since the images are $84 \times 84$ we used a different encoder and decoder, given by:

1. $4 \times 4$ conv. 16 ELU. stride 2. layer norm
2. $4 \times 4$ conv. 32 ELU. stride 2. layer norm
3. $4 \times 4$ conv. 32 ELU. stride 2. layer norm
4. $4 \times 4$ conv. 32 ELU. stride 2. layer norm
5. fully connected. 512 ELU. layer norm
6. recurrent. 250 Sigmoid. layer norm on the output
7. fully connected. 512 RELU. layer norm
8. fully connected. $8 \times 8 \times 64$ RELU. layer norm
9. $4 \times 4$ reshape 2 nearest-neighbour, conv. 32 RELU. layer norm
10. $4 \times 4$ reshape 2 nearest-neighbour, conv. 32 RELU. layer norm
11. $4 \times 4$ reshape 2 nearest-neighbour, conv. 16 RELU. layer norm
12. $4 \times 4$ reshape 2 nearest-neighbour, conv. 1 Sigmoid

We used the same architecture for $\Upsilon^{\text{R-NEM}}$, with the only difference that at each time-step we concatenated an embedding of the action produced by the agent to the hidden state. Here we used a single layer MLP with 10 units and a *ReLU* activation function to compute this embedding.

In the Atari experiment we trained with $K = 4$ and reduced the input noise to 0.02, in order to preserve tiny elements such as bullets (that only occupy 1-2 pixels).

