# OpenReview forum: "Relational Neural Expectation Maximization: Unsupervised Discovery of Objects and their Interactions"
_ICLR.cc/2018/Conference — Accept (Poster)_

### Official Review · AnonReviewer1 · 2017-11-28
**Clear contribution to neural intuitive physics models**

**Rating:** 7
**Confidence:** 4

**Review:**

Summary
---
This work applies a representaion learning technique that segments entities to learn simple 2d intuitive physics without per-entity supervision. It adds a relational mechanism to Neural Expectation Maximization and shows that this mechanism provides a better simulation of bouncing balls in a synthetic environment.

Neural Expectation Maximization (NEM) decomposes an image into K latent variables (vectors of reals) theta_k. A decoder network reconstructs K images from each of these latent variables and these K images are combined into a single reconstruction using pixel-wise mixture components that place more weight on pixels that match the ground truth. An encoder network f_enc() then updates the latent variables to better explain the reconstructions they produced.
The neural nets are learned so that the latent variables reconstruct the image well when used by the mixture model and match a prior otherwise. Previously NEM has been shown to learn variables which represent individual objects (simple shapes) in a compositional manner, using one variable per object.

Other recent neural models can learn to simulate simple 2d physics environments (balls bouncing around in a 2d plane). That work supervises the representation for each entity (ball) explicitly using states (e.g. position and velocity of balls) which are known from the physics simulator used to generate the training data. The key feature of these models is the use of a pairwise embedding of an object and its neighbors (message passing) to predict the object's next state in the simulation.

This paper paper combines the two methods to create Relational Neural Expectation Maximization (R-NEM), allowing direct interaction at inference time between the latent variables that encode a scene. The encoder network from NEM can be seen as a recurrent network which takes one latent variable theta_k at time t and some input x to produce the next latent variable theta_k at time t+1. R-NEM adds a relational module which computes an embedding used as a third input to the recurrent encoder. Like previous relational models, this one uses a pairwise embedding of the object being updated (object k) and its neighbors. Unlike previous neural physics models, R-NEM uses a soft attention mechanism to determine which objects are neighbors and which are not. Also unlike previous neural models, this method does not require per-object supervision.

Experiments show that R-NEM learns compositional representations that support intuitive physics more effectively than ablative baselines. These experiments
show:
1) R-NEM reconstructs images more accurately than baselines (RNN/LSTM) and NEM (without object interaction).
2) R-NEM is trained with 4 objects per image. It does a bit worse at reconstructing images with 6-8 objects per image, but still performs better than baselines.
3) A version of R-NEM without neighborhood attention in the relation module matches the performance of R-NEM using 4 objects and performs worse than R-NEM at 6-8 objects.
4) R-NEM learns representations which factorize into one latent variable per object as measured by the Adjusted Rand Index, which compares NEM's pixel clustering to a ground truth clustering with one cluster per object.
5) Qualitative and quantitative results show that R-NEM can simulate 2d ball physics for many time steps more effectively than an RNN and while only suffering gradual divergence from the ground truth simulation.

Qualitative results show that the attentional mechanism attends to objects which are close to the context object together, acting like the heuristic neighborhood mechanism from previous work.

Follow up experiments extend the basic setup significantly. One experiment shows that R-NEM demonstrates object permanence by correctly tracking a collision when one of the objects is completely occluded. Another experiment applies the method to the Space Invaders Atari game, showing that it treats columns of aliens as entities. This representation aligns with the game's goal.


Strengths
---

The paper presents a clear, convincing, and well illustrated story.

Weaknesses
---

* RNN-EM BCE results are missing from the simulation plot (right of figure 4).

Minor comments/concerns:

* 2nd paragraph in section 4: Are parameters shared between these 3 MLPs (enc,emb,eff)? I guess not, but this is ambiguous.

* When R-NEM is tested against 6-8 balls is K set to the number of balls plus 1? How does performance vary with the number of objects?

* Previous methods report performance across simulations of a variety of physical phenomena (e.g., see "Visual Interaction Networks"). It seems that supervision isn't needed for bouncing ball physics, but I wonder if this is the case for other kinds of phenomena (e.g., springs in the VIN paper). Can this method eliminate the need for per-entity supervision in this domain?

* A follow up to the previous comment: Could a supervised baseline that uses per-entity state supervision and neural message passsing (like the NPE from Chang et. al.) be included?

* It's a bit hard to qualitatively judge the quality of the simulations without videos to look at. Could videos of simulations be uploaded (e.g., via anonymous google drive folder as in "Visual Interaction Networks")?

* This uses a neural message passing mechanism like those of Chang et. al. and Battaglia et. al. It would be nice to see a citation to neural message passing outside of the physics simulation domain (e.g. to "Neural Message Passing for Quantum Chemistry" by Gilmer et. al. in ICML17).

* Some work uses neighborhood attention coefficients for neural message passing. It would be nice to see a citation included.
    * See "Neighborhood Attention" in "One-Shot Imitation Learning" by Duan et. al. in NIPS17
    * Also see "Programmable Agents" by Denil et. al.


Final Evaluation
---

This paper clearly advances the body of work on neural intuitive physics by incorporating NEM entity representation to allow for less supervision. Alternatively, it adds a message passing mechanism to the NEM entity representation technique. These are moderately novel contributions and there are only minor weaknesses, so this is a clear accept.

---

> ### Author Response · Authors · 2017-12-21
> **Reply to Reviewer 1 on how we will address the comments**
>
> Thank you for the careful consideration of our paper and for the useful feedback. Regarding your comments:
>
> - We will incorporate the RNN-EM BCE results in the simulation plot (right of figure 4).
> - Videos of the performance of R-NEM (compared to the RNN) on all balls tasks are available at https://sites.google.com/view/r-nem-gifs/home
> - We will incorporate related work on neural message passing as you suggested
>
> Indeed the parameters between the 3 MLPs (enc, emb, eff) are not shared and we will clarify this in the text.
>
> When R-NEM is tested against 6-8 balls we set K to 8. We have tried K=9 also and obtained similar results. In general we observe (as is in line with the findings in the Neural Expectation Maximization paper) that increasing K benefits performance (independent of the number of balls) at training and at test time. Very large values of K > 9-10 cause instabilities during training as there may be too many components competing for the same pixel in the E-step. Choosing K < # objects hinders performance, as expected. We are in the process of training R-NEM with K=8 on the balls 4 dataset to provide further insight into this.
>
> We expect R-NEM to be able to fully eliminate the need for per-entity state supervision in various other domains that revolve around interactions between entities. The interaction function incorporated in N-EM is not restricted to local interaction and therefore there is no apparent reason for it not to be able to handle springs.
>
> Although we are happy to include a supervised baseline, we have not been able to come up yet with a fair comparison measure. None of the supervised methods reconstructs in pixel-space and therefore comparing in terms of BCE would put R-NEM at a significant disadvantage. Being unable to disentangle error due to poorly modeling physical dynamics (including interactions) from error due to poor visual reconstruction only allow for comparing to approaches that reconstruct in pixel-space. We are currently looking into incorporating a stronger unsupervised baseline in the form of PredNet (https://arxiv.org/pdf/1605.08104.pdf).

---

### Official Review · AnonReviewer2 · 2017-11-29
**Interesting paper**

**Rating:** 7
**Confidence:** 3

**Review:**

This was a pretty interesting read. Thanks. A few comments:

One sentence that got me confused is this: “Unlike other work (Battaglia et al., 2016) this function is not commutative and we opt for a clear separation between the focus object k and the context object i as in previous work (Chang et al., 2016)”. What exactly is not commutative? The formulation seems completely align with the work of Battaglia et al, with the difference that one additionally has an attention on which edges should be considered (attention on effects). What is the difference to Battaglia et al. that this should highlight?

I don’t think is very explicit what k is in the experiments with bouncing balls. Is it 5 in all of them ? When running with 6-8 balls, how are balls grouped together to form just 5 objects?

Is there any chance of releasing the code/ data used in this experiments?

---

> ### Author Response · Authors · 2017-12-21
> **Reply to Reviewer 2 on how we will address the comments**
>
> Thank you for the careful consideration of our paper and for the useful feedback. Regarding your comments:
>
> We were mistaken in that we thought that the interaction function in Battaglia et al. 2016 was commutative in that it did not distinguish a sender and a receiver in computing interactions between objects. However upon reviewing their work again it turns out that that segment (https://arxiv.org/pdf/1612.00222.pdf page 3 - bottom) only referred to the ordering of the arguments in the a-function. We remove this part in the new draft.
>
> The number of components is K=5 for the 4 balls experiment, K=8 for the 678 balls experiment, and K=5 for the occlusion experiment. In general we observe (as is in line with the findings in the Neural Expectation Maximization paper) that increasing K benefits performance (independent of the number of balls) at training and at test time. Very large values of K > 9-10 cause instabilities during training as there may be too many components competing for the same pixel in the E-step. Choosing K < # objects hinders performance, as expected. We are in the process of training R-NEM with K=8 on the balls 4 dataset to provide further insight into this.
>
> Our current codebase is an adaptation of the code provided by the authors of the Neural Expectation Maximization paper (found here: https://github.com/sjoerdvansteenkiste/Neural-EM). We will release our adaptation of this code including all datasets upon publication.

---

### Official Review · AnonReviewer3 · 2017-11-30
**Very interesting work and the proposed approach is well explained. The experimental section could be improved.**

**Rating:** 8
**Confidence:** 5

**Review:**

Summary:
The manuscript extends the Neural Expectation Maximization framework by integrating an interaction function that allows asymmetric pairwise effects between objects. The network is demonstrated to learn compositional object representations which group together pixels, optimizing a predictive coding objective. The effectiveness of the approach is demonstrated on bouncing balls sequences and gameplay videos from Space Invaders. The proposed R-NEM model generalizes

Review:
Very interesting work and the proposed approach is well explained. The experimental section could be improved.
I have a few questions/comments:
1) Some limitations could have been discussed, e.g. how would the model perform on sequences involving more complicated deformations of objects than in the Space Invaders experiment? As you always take the first frame of the 4-frame stacks in the data set, do the objects deform at all?
2) It would have been interesting to vary K, e.g. study the behaviour for K in {1,5,10,25,50}. In Space Invaders the model would probably really group together separate objects. What happens if you train with K=8 on sequences of 4 balls and then run on 8-ball sequences instead of providing (approximately) the right number of components both at training and test time (in the extrapolation experiment).
3) One work that should be mentioned in the related work section is Michalski et al. (2014), which also uses noise and predictive coding to model sequences of bouncing balls and NORBvideos. Their model uses a factorization that also discovers relations between components of the frames, but in contrast to R-NEM the components overlap.
4) A quantitative evaluation of the bouncing balls with curtain and Space Invaders experiments would be useful for comparison.
5) I think the hyperparameters of the RNN and LSTM are missing from the manuscript. Did you perform any hyperparameter optimization on these models?
6) Stronger baselines would improve the experimental section, maybe Seo et al (2016). Alternatively, you could train the model on Moving MNIST (Srivastava et al., 2015) and compare with other published results.

I would consider increasing the score, if at least some of the above points are sufficiently addressed.

References:
Michalski, Vincent, Roland Memisevic, and Kishore Konda. "Modeling deep temporal dependencies with recurrent grammar cells""." In Advances in neural information processing systems, pp. 1925-1933. 2014.
Seo, Youngjoo, Michaël Defferrard, Pierre Vandergheynst, and Xavier Bresson. "Structured sequence modeling with graph convolutional recurrent networks." arXiv preprint arXiv:1612.07659 (2016).
Srivastava, Nitish, Elman Mansimov, and Ruslan Salakhudinov. "Unsupervised learning of video representations using lstms." In International Conference on Machine Learning, pp. 843-852. 2015.

---

> ### Author Response · Authors · 2017-12-21
> **Reply to Reviewer 3 on how we will address the comments**
>
> Thank you for the careful consideration of our paper and for the useful feedback. Regarding your comments:
>
> 1) We agree that adding a general discussion on the limitations of our approach is valuable (and currently lacking) and we intend to include this in the new draft. In response to your specific examples: we expect the performance of R-NEM to be similar in regard to N-EM when confronted with high variability within an object class. In N-EM when trained on sequences of moving MNIST digits (that highly vary, since each MNIST digit is unique) it is more difficult to group pixels into coherent objects (digits in this case) as all variation needs to be captured. However, once pixels have been clustered to belong to an object, and its deformation is consistent/predictable R-NEM should be able to accurately capture it (as is for example the case for the occlusion experiment). If it is not consistent/predictable as is for example the case in Atari due to randomness and deformation due to down-sampling, then R-NEM will continuously try to adjust its predictions as a function of feedback from the environment (eg. by means of the masked difference between prediction and reality that is fed into the system) .
>
> 2) We have tried a range of values for K={4,5,6,7,8} on Atari, before settling onto K=4. Since the Aliens move together they are mostly grouped together into a single component, which leaves only bullets / shields and the space-ship as remaining objects. The bullets from the Aliens can not be predicted and so usually one of the alien columns takes care of it. Bullets from the Spaceship can be predicted (as we feed in the action) and for this a remaining group does help. It should be noted that (similar to N-EM) very large values of K > 9-10 cause instabilities during training as there may be too many components competing for the same pixel in the E-step. We have therefore not explored extreme cases of K=25/50 further. We are in the process of training R-NEM with K=8 on the balls 4 dataset to provide further insight into its behavior of 4-balls and 678-balls.
>
> 3) Thank you, we’ve missed this relevant work and will include it in the next draft.
>
> 4) We are in the process of training RNN / LSTM models on the bouncing balls with curtain task to provide a quantitative evaluation. With regard to a quantitative evaluation on Atari we feel that it would take an unjustified/disproportionate amount of effort and computational time to  provide a comparison that carries any value. In particular, since we have not studied the performance of RNN / LSTM on this domain previously, we would need to perform a general search to ensure that our baseline isn’t just underfitted.
>
> 5) We use the same encoder / decoder architecture for the LSTM / RNN variations, that also receive the difference between prediction from the previous timestep and current frame as input. We use the same layer size also (250 units). This is reported in the Appendix. Some experiments that we ran but did not report involve adding more units (500) for the LSTM / RNN, for which we did not observe significant differences in performance. Moreover, we experimented with an RHN, but were unable to improve upon the LSTM and therefore left it out of the final comparison. We have tried several deeper/wider variations of the interaction function and settled on the smallest architecture that was able to achieve the reported performance. This last observation is mentioned in the Appendix.
>
> 6) We are happy to incorporate stronger baselines in our quantitative evaluation of R-NEM. However we are unsure whether Seo et al. (2016) / Srivastava et al. 2015 (2015) are suitable. Both approaches are encoder / decoder architectures that first encode a sequence of time-steps in an LSTM, to then use a (or multiple) decoder LSTMs initialized with the encoded state to reconstruct the input sequence in reverse, or predict future time-steps. Since our model is trained with next-step prediction, this reduces the Decoder LSTM of such an approach to a simple feedforward decoder. Moreover, we are only computing gradients from next-step prediction, eliminating the reverse decoder LSTM from such an approach such that the model is essentially reduced to a standard LSTM again that we do compare to. In case of Seo et al. (2016) the only addition would be to use a graph-convolutional LSTM in stead. Neither of these seem much stronger than our LSTM baseline on the balls environments. Instead we would like to propose to use PredNet (Lotter et al. https://arxiv.org/pdf/1605.08104.pdf) as an alternative to your suggestions. In response to your suggestion to evaluate on the moving MNIST dataset we would like to point out that this dataset does involve any interactions between the digits and is unsuitable to evaluate the impact of our interaction function.

---

> > ### Comment · AnonReviewer3 · 2018-01-12
> > **Increased score to 8**
> >
> > The rebuttal and revision addressed enough of my concerns for me to increase the score to 8.
> > Good work on the additional experiments and the discussion of limitations in the conclusion!

---

### Author Response · Authors · 2017-12-21
**Update paper title**

We updated the title of our paper to better reflect the nature of our approach

---

### Author Response · Authors · 2018-01-05
**Changes to draft in response to reviewer comments**

We have updated the draft of the paper to reflect the comments of the reviewers and incorporated the following changes:

- Results for R-NEM with K=8 on the bouncing balls task
- Loss comparison of R-NEM to RNN and LSTM on the curtain task
- Link to videos of all balls experiments: https://sites.google.com/view/r-nem-gifs
- Clarification of experimental set-up
- Simulation results of RNN-EM to figure 4
- Discussion on the limitations of our approach
- Suggested references
- General changes to improve readability

We were unable to incorporate a stronger unsupervised baseline (PredNet) as of yet. This is still something that we are looking into.

---

### Decision · Program_Chairs · 2018-01-29
**ICLR 2018 Conference Acceptance Decision**

**Decision:**

Accept (Poster)

**Comment:**

All three reviewers recommend acceptance. The authors did a good job at the rebuttal which swayed the first reviewer to increase the final rating. This is a clear accept.